# Steer LLM Latents for Hallucination Detection

**Seongheon Park** [1]  **Xuefeng Du** [1]  **Min-Hsuan Yeh** [1]  **Haobo Wang** [2]  **Yixuan Li** [1]

## Abstract

Hallucinations in LLMs pose a significant concern to their safe deployment in real-world applications. Recent approaches have leveraged the latent space of LLMs for hallucination detection, but their embeddings, optimized for linguistic coherence rather than factual accuracy, often fail to clearly separate truthful and hallucinated content. To this end, we propose the **T**ruthfulness **S**eparator **V**ector (**TSV**), a lightweight and flexible steering vector that reshapes the LLM's representation space during inference to enhance the separation between truthful and hallucinated outputs, without altering model parameters. Our two-stage framework first trains TSV on a small set of labeled exemplars to form compact and well-separated clusters. It then augments the exemplar set with unlabeled LLM generations, employing an optimal transport-based algorithm for pseudo-labeling combined with a confidence-based filtering process. Extensive experiments demonstrate that TSV achieves state-of-the-art performance with minimal labeled data, exhibiting strong generalization across datasets and providing a practical solution for real-world LLM applications.

## 1. Introduction

Large language models (LLMs) have demonstrated remarkable capabilities in natural language understanding and generation, showcasing their potential as general-purpose task solvers (Zhao et al., 2023). Despite their success, LLMs can generate hallucinated outputs—statements that appear plausible but factually inaccurate or unsupported. Such hallucinations can undermine user trust and lead to potentially harmful consequences, especially in high-stake applications (Zhang et al., 2023; Pal et al., 2023). Therefore, to be truly trustworthy, an LLM must not only generate text that is

[1]Department of Computer Sciences, University of Wisconsin-Madison [2]School of Software Technology, Zhejiang University. Correspondence to: Yixuan Li <sharonli@cs.wisc.edu>.

*Proceedings of the 42nd International Conference on Machine Learning*, Vancouver, Canada. PMLR 267, 2025. Copyright 2025 by the author(s).

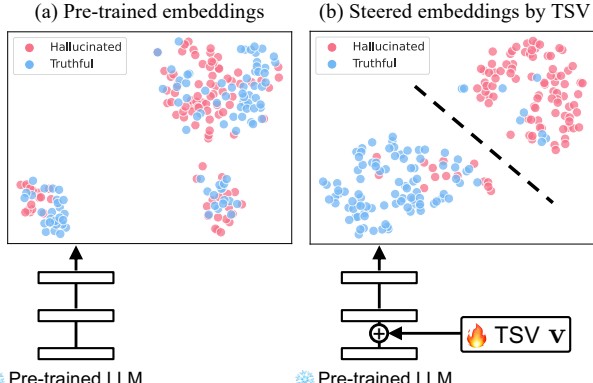

*Figure 1.* T-SNE visualization (Van der Maaten & Hinton, 2008) of the last-token embeddings from the final layer of LLaMA-3.1-8B on the TruthfulQA test set. (a) Pre-trained model's embeddings exhibit significant overlap, whereas (b) adding TSV to latent states of an intermediate LLM layer effectively separates the embeddings of truthful and hallucinated data.

consistent with user prompts but also possess the ability to detect hallucinations and alert users when they occur.

Recent work has explored leveraging the latent space of LLMs to identify hallucinations (Burns et al., 2023; Azaria & Mitchell, 2023; Marks & Tegmark, 2024; Yin et al., 2024a; Du et al., 2024; Chen et al., 2024a; Li et al., 2024; Kossen et al., 2024; Orgad et al., 2025). These approaches typically rely on the embeddings of off-the-shelf LLMs to classify outputs as truthful or hallucinated. However, pre-trained LLMs are optimized for linguistic coherence using a next-token prediction objective, often prioritizing fluency and syntactic correctness over factual accuracy (Radford et al., 2019). As a result, their internal representations, while powerful for general text generation, can fail to provide a clear separation between truthful and hallucinated content (see real-world example in Figure 1a). This motivates a key question:

> *How can we shape the latent space of an LLM for hallucination detection?*

Instead of fine-tuning the LLMs, which is computationally expensive and alters the model's parameters (Gekhman et al., 2024), we propose learning a lightweight vector, called **T**ruthfulness **S**eparator **V**ector (**TSV**). As illustrated in Figure 1b, this learnable vector is introduced during inference

and adjusting the internal representations of the LLM to enhance the separation between truthful and hallucinated generations, without modifying the model's parameters. TSV focuses on reshaping the latent space for *classifying* hallucinated responses, a fundamentally different objective from mitigating hallucinated generations (Li et al., 2024; Chen et al., 2024b; Marks & Tegmark, 2024). To the best of our knowledge, this is the first exploration of steering representations for hallucination detection.

Learning TSV is appealing yet challenging due to the lack of large-scale human-labeled datasets with truthfulness annotations for LLM generation, which are costly and time-intensive to create. To overcome this, we propose a two-stage training framework. In the initial stage, a small exemplar set of labeled data is used to guide the learning process. The objective in this stage is to encourage the steered embeddings to form compact clusters around class prototypes, representing truthful and hallucinated generations. In the second stage, we augment the training data by leveraging unlabeled LLM generations, which are freely available for deployed LLM systems through user queries and interactions (Du et al., 2024). To assign pseudo-labels to these unlabeled samples, we propose an optimal transport-based algorithm, which aligns unlabeled data embeddings with class prototypes by minimizing transport costs while accounting for the imbalanced class proportions. Furthermore, a confidence-based sample selection is then used to include only the most reliable pseudo-labeled samples in the training process. Together, these stages enable TSV to effectively separate truthful and hallucinated representations while significantly reducing the reliance on human labeling.

Extensive experiments demonstrate the strong performance of our method across diverse datasets. On the challenging TruthfulQA benchmark (Lin et al., 2022a), our approach achieves a significant **+12.8%** improvement in hallucination detection accuracy (AUROC) compared to state-of-the-art methods. Notably, our method reaches performance comparable to the fully-supervised upper bound (*e.g.*, 84.2% vs. 85.5% on TruthfulQA), while using a small labeled exemplar set with only 32 examples. TSV also exhibits strong generalization capabilities, maintaining competitive performance when applied to unseen datasets. Our key contributions are summarized as follows:

1. We propose the Truthfulness Separator Vector (TSV), a lightweight approach to separate truthful and hallucinated representations without fine-tuning the LLMs, which is largely unexplored in hallucination detection.

2. We develop an optimal transport-based pseudo-labeling framework with confidence-based sample selection to leverage unlabeled LLM generations effectively.

3. We demonstrate TSV's superior performance and perform in-depth ablation studies to evaluate the impact of various design choices in TSV and validate its scalability across larger LLMs and diverse datasets. These findings provide a systematic understanding of leveraging steering vector and limited labeled data for hallucination detection, paving the way for future research.

## 2. Related Works

**Hallucination detection** has emerged as a critical area of research, addressing safety concerns of LLMs and their deployment in real-world applications (Huang et al., 2023). A plethora of works address hallucination detection by designing uncertainty scoring functions. For instance, logit-based methods utilize token-level probability as an uncertainty score (Ren et al., 2022; Malinin & Gales, 2021; Kuhn et al., 2023), verbalized methods prompt LLMs to express their uncertainty in human language (Lin et al., 2022b; Xiong et al., 2024), and consistency-based methods assess uncertainty by evaluating the consistency across multiple responses (Manakul et al., 2023; Chen et al., 2024a). Recently, internal state-based methods such as HaloScope leverage hidden activations to identify hallucination subspace (Du et al., 2024). However, these approaches often rely on default LLM embeddings that *do not inherently separate truthful and hallucinated data*. In contrast, our method aims to shape the latent space through a learnable steering vector for enhanced separation between the two types of data.

On the other hand, supervised methods leverage labeled data to train the classifier, assuming that pre-trained LLMs encode the truthfulness of responses within their internal states (Azaria & Mitchell, 2023; Marks & Tegmark, 2024). However, these methods require extensive labeling efforts. In contrast, our method performs hallucination detection with minimal human supervision, which is more practical for real-world applications.

**Activation engineering** enables control over the LLM generation during inference, applying task-specific steering vectors into the model's internal representation (Im & Li, 2025). For example, several studies mitigate hallucination by shifting activations along the truthful direction identified by analyzing activation differences between contrastive pairs (Li et al., 2024; Chen et al., 2024b; Marks & Tegmark, 2024). Concurrently, representation fine-tuning methods introduce learning task-specific interventions on linear subspaces of hidden representations (Wu et al., 2024) or sparse subsets of attention heads (Yin et al., 2024b).

Our approach differs in the following key aspects: (1) We learn a steering vector specifically for hallucination detection, focusing on separating representations rather than mitigating hallucinated generations, and (2) while previous methods rely on large labeled datasets, our method achieves strong performance under minimal human supervision.

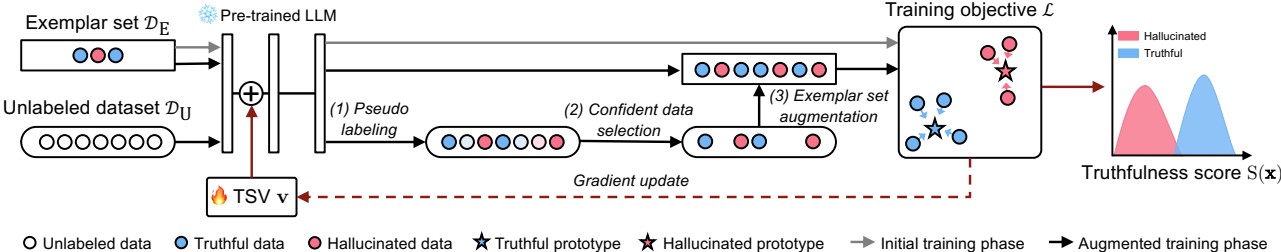

*Figure 2.* Overall framework. In the initial training phase, Truthfulness Separator Vector (TSV) is trained on an exemplar set. After initial training, (1) we assign soft pseudo-labels to the unlabeled data, (2) select confident pseudo-labeled samples, and (3) augment the exemplar set with selected samples. Finally, we retrain TSV with the augmented set. Best viewed in color.

## 3. Problem Setup

**Definition 3.1** (**Hallucination detector**). *We define the truthful distribution $\mathbb{P}_{true}$ as the joint distribution over pairs of the input prompts and their corresponding truthful generations. Let $\mathcal{V}$ denote a vocabulary space of a causal LLM, where each individual token is denoted as $x \in \mathcal{V}$. Given an input prompt $\mathbf{x}_{prompt} = (x_1, \ldots, x_n)$ and a model generation $\tilde{\mathbf{x}} = (x_{n+1}, \ldots, x_{n+m})$, the task of hallucination detection aims to learn a binary predictor $G : \mathcal{X} \to \{0, 1\}$:*

$$G(\mathbf{x}_{prompt}, \tilde{\mathbf{x}}) = \begin{cases} 1, & if \, (\mathbf{x}_{prompt} \oplus \tilde{\mathbf{x}}) \sim \mathbb{P}_{true} \\ 0, & otherwise \end{cases}, \quad (1)$$

*where $\mathbf{x}_{prompt} \oplus \tilde{\mathbf{x}} = (x_1, \ldots, x_n, x_{n+1}, \ldots, x_{n+m})$ represents the ordered concatenation of the prompt $\mathbf{x}_{prompt}$ and the generation $\tilde{\mathbf{x}}$.*

Following the practical setup in recent work HaloScope (Du et al., 2024), we utilize unlabeled LLM generations in the wild, which can be collected in vast quantities through user interactions with LLMs. This data can be freely collected for any deployed LLM system, yet often contains a mixture of truthful and hallucinated content. Formally,

**Definition 3.2** (**Unlabeled data**). *We define the unlabeled pairs of input prompt $\mathbf{x}_{prompt}^i$ and LLM generation in the wild $\tilde{\mathbf{x}}^i$ to be the following mixture of distributions:*

$$\mathbb{P}_{unlabeled} = (1 - \pi)\mathbb{P}_{true} + \pi\mathbb{P}_{hallucination},$$

*where $\pi \in [0, 1]$ is the fraction of hallucinated generation.*

The unlabeled dataset, $\mathcal{D}_U = \{(\mathbf{x}_{prompt}^1 \oplus \tilde{\mathbf{x}}^1), \ldots, (\mathbf{x}_{prompt}^M \oplus \tilde{\mathbf{x}}^M)\}$, is independently and identically sampled from the mixture distribution $\mathbb{P}_{unlabeled}$. Here, $M$ is the total number of unlabeled samples, and the tilde symbolizes the uncertain nature of the generation.

**Exemplar set.** In addition to the unlabeled data, we incorporate a small, practical-to-annotate set of labeled exemplars to guide the learning of hallucination detector. Specifically, pairs of input prompt $\mathbf{e}_{prompt}^i$ and LLM-generated

responses $\tilde{\mathbf{e}}^i$ can be annotated with ground-truth labels $c_i \in \mathcal{C} = \{\text{truthful}, \text{hallucinated}\}$. This forms the labeled exemplar set: $\mathcal{D}_E = \{(\mathbf{e}_{prompt}^1 \oplus \tilde{\mathbf{e}}^1, c_1), \ldots, (\mathbf{e}_{prompt}^N \oplus \tilde{\mathbf{e}}^N, c_N)\}$, where $N$ is the total number of labeled exemplars. In this paper, we will show that $N$ can be kept very small (e.g., 32) to minimize annotation costs while still providing valuable guidance for the learning process.

## 4. Method

**Overview.** Since LLMs do not inherently produce optimal embeddings to separate truthful and hallucinated data, our framework introduces a learnable vector, named **T**ruthfulness **S**eparator **V**ector (**TSV**), designed to enhance this separation within the representation space of the LLM. As illustrated in Figure 2, TSV is added into the latent states of the model during inference, which avoids the computational overhead associated with retraining or fine-tuning the model. In what follows, we describe how to learn TSV using unlabeled data and a small exemplar set.

### 4.1. How to learn TSV? Initial training phase

TSV is defined as a single trainable vector $\mathbf{v} \in \mathbb{R}^d$, which can be plugged into pre-trained LLMs after the generation is completed—**without** compromising their original language capabilities. Given a sequence of tokens (*e.g.*, input prompt and generation pair), we add $\mathbf{v}$ to $\mathbf{h}^{(l)}$, which represent the $d$-dimensional latent states at an intermediate layer $l$:

$$\mathbf{h}^{(l)} \leftarrow \mathbf{h}^{(l)} + \lambda\mathbf{v}, \quad (2)$$

where $\lambda$ is a hyperparameter which controls the strength of the steering, and $\mathbf{v}$ is shared across all token positions. This intervention affects the embeddings in subsequent layers $l + 1, \ldots, L$ via the non-linear transformations inherent in LLM architecture. The last-token embedding at the final layer after applying TSV is:

$$\Phi_{\text{final}}(\mathbf{h}^{(l)} + \lambda\mathbf{v}) = \phi_L \circ \phi_{L-1}... \circ \phi_{l+1}(\mathbf{h}^{(l)} + \lambda\mathbf{v}),$$

where $\phi_l$ indicates the non-linear transformation in layer $l$ of the transformer model. In Section 5.3, we perform ablations on different layers of applying TSV.

**Training objective of TSV.** To effectively detect hallucinations, it is crucial to establish a clear decision boundary between truthful and hallucinated data. To this end, we propose a training objective that learns TSV to separate embeddings between two classes $\mathcal{C} = \{\text{truthful}, \text{hallucinated}\}$. This is achieved by performing maximum likelihood estimation (MLE) on the exemplar set $\mathcal{D}_{\mathrm{E}}$:

$$\arg\max_{\mathbf{v}} \prod_{i=1}^{|\mathcal{D}_{\mathrm{E}}|} p(c_i \mid \Phi_{\text{final}}(\mathbf{h}_i^{(l)} + \lambda\mathbf{v})), \tag{3}$$

where $i$ is the index of training sample in $\mathcal{D}_{\mathrm{E}}$.

To realize the MLE objective, we need to explicitly model the probability distribution $p(c_i \mid \Phi_{\text{final}}(\mathbf{h}_i^{(l)} + \lambda\mathbf{v}))$. In particular, we model the last-token embeddings at the final layer using a hyperspherical distribution with the unit norm, where truthful and hallucinated data each form distinct clusters. This modeling aligns with the structure of embeddings typically observed after the RMSNorm layer in practical Transformer models (Dubey et al., 2024; Yang et al., 2024), where the norms of the embeddings are similar but directions can vary (see verification in **Appendix G**). This can be naturally characterized by the von Mises-Fisher distribution, a classical probability distribution in directional statistics (Mardia & Jupp, 2009), which is analogous to spherical Gaussian distributions for features with unit norms. Under this model, the class conditional probability is given by:

$$p(c \mid \mathbf{r}^{\mathbf{v}}) = \frac{\exp\left(\kappa \boldsymbol{\mu}_c^{\top} \mathbf{r}^{\mathbf{v}}\right)}{\sum_{c'} \exp\left(\kappa \boldsymbol{\mu}_{c'}^{\top} \mathbf{r}^{\mathbf{v}}\right)}, \tag{4}$$

where $\mathbf{r}^{\mathbf{v}} = \Phi_{\text{final}}(\mathbf{h}^{(l)} + \lambda\mathbf{v})/\|\Phi_{\text{final}}(\mathbf{h}^{(l)} + \lambda\mathbf{v})\|_2$, represents the normalized last-token embedding at the final layer, $\boldsymbol{\mu}_c \in \mathbb{R}^d$ is the class prototype for class $c$, $\kappa \geq 0$ is the concentration parameter controlling how tightly the distribution is clustered around the mean direction $\boldsymbol{\mu}_c$.

**Empirical loss function.** Under the probability model defined above, our MLE objective in Eq. 3 is equivalent to minimizing the negative log-likelihood over the exemplar set $\mathcal{D}_{\mathrm{E}}$. This encourages embeddings within each class to cluster tightly around their respective class centroids:

$$\mathcal{L} = -\frac{1}{|\mathcal{D}_{\mathrm{E}}|} \sum_{i=1}^{|\mathcal{D}_{\mathrm{E}}|} \sum_{c \in \mathcal{C}} q(c \mid \mathbf{r}_i^{\mathbf{v}}) \log p(c \mid \mathbf{r}_i^{\mathbf{v}}) \tag{5}$$

where $q(\cdot \mid \mathbf{r}_i^{\mathbf{v}})$ denotes the target label distribution, which can be either ground-truth or pseudo-label.

**Prototype update.** In practice, the prototype vector $\boldsymbol{\mu}_c$ can be efficiently updated using exponential moving average (Wang et al., 2022):

$$\boldsymbol{\mu}_c \leftarrow \text{normalize}[\alpha\boldsymbol{\mu}_c + (1 - \alpha)\bar{\mathbf{r}}^{\mathbf{v}}], \tag{6}$$

where $\alpha$ is the decay rate, and $\bar{\mathbf{r}}^{\mathbf{v}} = \sum_i \frac{q(c|\mathbf{r}_i^{\mathbf{v}}) \cdot \mathbf{r}_i^{\mathbf{v}}}{\sum_j q(c|\mathbf{r}_j^{\mathbf{v}})}$ denotes the mean of the normalized embeddings from class $c$.

## 4.2. How to learn TSV? Augmented training phase

While we demonstrate that leveraging a few labeled examples helps hallucination detection (Section 5.3), these examples may not fully capture the diversity inherent in the truthful and hallucinated data distributions. To address the limitation, we propose to further incorporate unlabeled training data to augment the learning process.

**Label assignment via optimal transport.** Assigning labels (truthful vs. hallucinated) to unlabeled data is a non-trivial task, particularly because we aim to generate pseudo-labels that align with the class distribution of LLM generations, which are naturally imbalanced (Hu et al., 2024). To this end, we propose leveraging Optimal Transport (OT) (Villani et al., 2009), which provides a principled approach to label assignment. This approach aligns unlabeled data embeddings with class prototypes by minimizing transport costs while respecting the imbalanced class proportions. Given unlabeled dataset $\mathcal{D}_{\mathrm{U}}$ with $M$ samples, the optimization problem is formulated as:

$$\min_{\mathbf{Q} \in [0,1]^{M \times 2}} \quad -\sum_{m=1}^{M} \sum_{c \in \mathcal{C}} \mathbf{Q}_{m,c} \log \mathbf{P}_{m,c} - \epsilon H(\mathbf{Q})$$
$$\text{s.t.} \quad \mathbf{Q}\mathbf{1}_2 = \frac{1}{M}\mathbf{1}_M, \tag{7}$$
$$\mathbf{Q}^{\top}\mathbf{1}_M = \mathbf{w},$$

where $\mathbf{1}_M \in \mathbb{R}^M$ denotes an $M$-dimensional vector of ones, $\mathbf{Q}_{m,c} = \frac{1}{M}q(c|\mathbf{r}_m^{\mathbf{v}})$ represents an entry of the matrix $\mathbf{Q} \in \mathbb{R}^{M \times 2}$ for assigned joint pseudo-label probabilities, and $\mathbf{P}_{m,c} = \frac{1}{M}p(c|\mathbf{r}_m^{\mathbf{v}})$ denotes an entry of $\mathbf{P} \in \mathbb{R}^{M \times 2}$ for joint probabilities estimated by our model after initial training, where $p(c|\mathbf{r}_m^{\mathbf{v}})$ is computed with Eq. 4. The first constraint ensures that for each unlabeled sample, the total probability mass of being assigned to two classes adds up to 1. The second constraint ensures that the number of samples assigned to each class matches the expected class probability distribution $\mathbf{w} \in \mathbb{R}^2$. Here, $H(\mathbf{Q}) = -\sum_{ij} \mathbf{Q}_{ij} \log \mathbf{Q}_{ij}$ is the entropy function, and $\epsilon$ is a hyperparameter controlling the smoothness of the assignment, which we set to 0.05. The entropy regularization term enables the computationally efficient Sinkhorn algorithm (Cuturi, 2013) to solve the problem. The minimizer of Eq. 7 can be expressed as:

$$\mathbf{Q} = \text{diag}(\alpha)\mathbf{P}^{1/\epsilon}\text{diag}(\beta), \tag{8}$$

where $\alpha \in \mathbb{R}^M$ and $\beta \in \mathbb{R}^2$ are scaling coefficient vectors ensuring that the resulting $\mathbf{Q}$ forms a valid probability matrix. These scaling coefficients are determined iteratively using the following updates:

$$\alpha \leftarrow \frac{1}{M}\frac{\mathbf{1}_M}{\mathbf{P}^{1/\epsilon}\beta}, \quad \beta \leftarrow \frac{\mathbf{w}}{(\mathbf{P}^{1/\epsilon})^{\top}\alpha}. \tag{9}$$

We use the class distribution of the exemplar set as a proxy for $\mathbf{w}$, assuming a missing completely at random (MCAR) scenario, which is a natural assumption for data collected in real-world settings (Van Buuren, 2018).

**Confident data selection.** Since pseudo-labels predicted for the unlabeled data may be incorrect and thus introduce noise into the learning process, we propose selecting only the most "confident" pseudo-labeled samples from the unlabeled dataset $\mathcal{D}_\mathrm{U}$, which are most likely to be correct. We measure the model's predictive uncertainty using the cross-entropy between the assigned pseudo-label distribution $q$ and the model's predicted distribution $p$. Specifically, for each unlabeled sample $\mathbf{r}_i$, we define:

$$\Omega = \left\{ - \sum_{c \in \mathcal{C}} q(c \mid \mathbf{r}_i^\mathbf{v}) \log p(c \mid \mathbf{r}_i^\mathbf{v}) \,\middle|\, i \in \mathcal{I}_{\mathcal{D}_\mathrm{U}} \right\}, \quad (10)$$

where $\mathcal{I}_{\mathcal{D}_\mathrm{U}}$ denotes the index set of $\mathcal{D}_\mathrm{U}$. We then select $K$ samples from $\mathcal{D}_\mathrm{U}$ to form the subset $\mathcal{D}_\mathrm{S}$:

$$\mathcal{D}_\mathrm{S} = \{\mathcal{D}_\mathrm{U}^j \mid j \in \mathrm{TopK}_{i \in \mathcal{I}_{\mathcal{D}_\mathrm{U}}}(-\Omega_i)\}, \quad (11)$$

where TopK denotes the indices of the $K$ samples with the lowest uncertainty values, and $\mathcal{D}_\mathrm{U}^j$ is $j$-th data in $\mathcal{D}_\mathrm{U}$.

**Exemplar set augmentation.** Finally, we augment the original training dataset $\mathcal{D}_\mathrm{E}$ by incorporating the selected samples $\mathcal{D}_\mathrm{S}$ along with their pseudo-labels:

$$\mathcal{D}_\mathrm{E} \leftarrow \mathcal{D}_\mathrm{E} \cup \mathcal{D}_\mathrm{S}. \quad (12)$$

The learning process described in Section 4.1 is then repeated using the augmented dataset until convergence. We summarize the full algorithm in **Appendix A**.

### 4.3. Inference-time hallucination detection

During inference, we leverage the learned class prototypes $\boldsymbol{\mu}_c$ to perform hallucination detection. Specifically, we compute the truthfulness score as the normalized probability of a test input's embedding vector $\mathbf{r}_\mathrm{test}^\mathbf{v}$ being assigned to the truthful class. The scoring function is defined as:

$$S(\mathbf{x}') = \frac{\exp\left(\kappa \boldsymbol{\mu}_\mathrm{truthful}^\top \mathbf{r}_\mathrm{test}^\mathbf{v}\right)}{\sum_{c'} \exp\left(\kappa \boldsymbol{\mu}_{c'}^\top \mathbf{r}_\mathrm{test}^\mathbf{v}\right)}. \quad (13)$$

Based on the scoring function, the hallucination detector is $G_\zeta(\mathbf{x}_\mathrm{test}) = \mathbb{1}\{S(\mathbf{x}_\mathrm{test}) \geq \zeta\}$, where 1 indicates the truthful class and 0 indicates otherwise. The task can be seamlessly switched back to the original text generation by simply removing TSV $\mathbf{v}$, restoring the model's initial generation capabilities without additional modifications.

## 5. Experiments

### 5.1. Setup

**Datasets.** We evaluate our method on four generative question-answering (QA) tasks: three open-domain QA datasets–TruthfulQA (Lin et al., 2022a), TriviaQA (Joshi et al., 2017), and NQ Open (Kwiatkowski et al., 2019); and a domain-specific QA dataset–SciQ (Welbl et al., 2017). For evaluation, 25% of the QA pairs from each dataset are reserved for testing. Consistent with Du et al. (2024), 100 QA pairs are used for validation, while the remaining samples simulate the unlabeled training dataset. We randomly sample $N = 32$ pairs from TruthfuQA, and 64 pairs from the other datasets to construct an exemplar set, with $K = 128$ used for all experiments. Implementation details are provided in **Appendix B**.

**Models.** We evaluate our method using two families of widely adopted open-source LLMs which provide accessible internal representations: LLaMA-3.1-8b & 70b (Dubey et al., 2024), and Qwen-2.5-7b & 14b (Yang et al., 2024). By default, we used greedy sampling for the generation.

**Baselines.** We evaluate our approach against a diverse set of 11 baseline methods, including existing state-of-the-art. The baselines are categorized as follows: (1) logit-based methods–Perplexity (Ren et al., 2022), Length-Normalized Entropy (LN-entropy) (Malinin & Gales, 2021) and Semantic Entropy (Kuhn et al., 2023); (2) consistency-based methods–Lexical Similarity (Lin et al., 2024), Self-CKGPT (Manakul et al., 2023) and EigenScore (Chen et al., 2024a); (3) verbalized methods–Verbalize (Lin et al., 2022b) and Self-evaluation (Kadavath et al., 2022); and (4) internal state-based methods–Contrast-Consistent Search (CCS) (Burns et al., 2023), HaloScope (Du et al., 2024), and SAPLMA (Azaria & Mitchell, 2023). To ensure a fair comparison, all methods are evaluated on the same test dataset, using their default experimental configurations as specified in the respective literature.

**Evaluation.** Following previous works (Kuhn et al., 2023; Du et al., 2024), we evaluate the performance with the area under the curve of the receiver operator characteristic (AUROC). We consider the generation truthful when the similarity score between the generation and the reference answer is larger than a pre-defined threshold (e.g., 0.5). Following Lin et al. (2022a), we utilize BLEURT (Sellam et al., 2020) to measure the similarity. Additionally, we show that our method is robust when evaluated using GPT-4o (Hurst et al., 2024) in **Appendix E.2**.

### 5.2. Main results

In Table 1, we compare TSV with competitive hallucination detection methods from the literature. TSV demonstrates state-of-the-art performance, significantly outperforming other methods on both the LLaMA-3.1-8b and Qwen-2.5-7b models. We show that *unsupervised methods often struggle with inconsistent performance across different models and data distributions* as the representations in LLMs are not inherently aligned with the hallucination detection task,

*Table 1.* Main results. Comparison with competitive hallucination detection methods on different datasets. "Single sampling" indicates whether the approach requires multiple generations during inference. For our method, the mean and standard deviation are computed across three different random seeds. ♣ denotes methods trained on fully labeled datasets. All values are percentages (AUROC), and the best results are highlighted in **bold**.

| Model | Method | Single Sampling | TruthfulQA | TriviaQA | SciQ | NQ Open |
|---|---|---|---|---|---|---|
| LLaMA-3.1-8b | Perplexity | ✓ | 71.4 | 76.3 | 52.6 | 50.3 |
| | LN-Entropy | ✗ | 62.5 | 55.8 | 57.6 | 52.7 |
| | Semantic Entropy | ✗ | 59.4 | 68.7 | 68.2 | 60.7 |
| | Lexical Similarity | ✗ | 49.1 | 71.0 | 61.0 | 60.9 |
| | EigenScore | ✗ | 45.3 | 69.1 | 59.6 | 56.7 |
| | SelfCKGPT | ✗ | 57.0 | 80.2 | 67.9 | 60.0 |
| | Verbalize | ✓ | 50.4 | 51.1 | 53.4 | 50.7 |
| | Self-evaluation | ✓ | 67.8 | 50.9 | 54.6 | 52.2 |
| | CCS | ✓ | 66.4 | 60.1 | 77.1 | 62.6 |
| | HaloScope | ✓ | 70.6 | 76.2 | 76.1 | 62.7 |
| | SAPLMA♣ | ✓ | 78.2 | 83.7 | 77.3 | 62.8 |
| | **TSV (Ours)** | ✓ | **84.2**$^{\pm0.2}$ | **84.0**$^{\pm0.5}$ | **85.8**$^{\pm0.4}$ | **76.1**$^{\pm0.7}$ |
| | **TSV♣ (Ours)** | ✓ | **85.5**$^{\pm0.1}$ | **87.2**$^{\pm0.2}$ | **88.6**$^{\pm0.1}$ | **78.0**$^{\pm0.2}$ |
| Qwen-2.5-7b | Perplexity | ✓ | 65.1 | 50.2 | 53.4 | 51.2 |
| | LN-Entropy | ✗ | 66.7 | 51.1 | 52.4 | 54.3 |
| | Semantic Entropy | ✗ | 66.1 | 58.7 | 65.9 | 65.3 |
| | Lexical Similarity | ✗ | 49.0 | 63.1 | 62.2 | 61.2 |
| | EigenScore | ✗ | 53.7 | 61.3 | 63.2 | 57.4 |
| | SelfCKGPT | ✗ | 61.7 | 62.3 | 58.6 | 63.4 |
| | Verbalize | ✓ | 60.0 | 54.3 | 51.2 | 51.2 |
| | Self-evaluation | ✓ | 73.7 | 50.9 | 53.8 | 52.4 |
| | CCS | ✓ | 67.9 | 53.0 | 51.9 | 51.2 |
| | HaloScope | ✓ | 81.3 | 73.4 | 76.6 | 65.7 |
| | SAPLMA♣ | ✓ | 81.7 | **82.0** | 81.5 | 67.9 |
| | **TSV (Ours)** | ✓ | **87.3**$^{\pm0.4}$ | 79.8$^{\pm0.9}$ | **82.0**$^{\pm0.4}$ | **73.8**$^{\pm0.7}$ |
| | **TSV♣ (Ours)** | ✓ | **88.7**$^{\pm0.1}$ | **84.2**$^{\pm0.5}$ | **84.8**$^{\pm0.3}$ | **76.2**$^{\pm0.3}$ |

making them less reliable for safety-critical applications. In contrast, our method achieves robust and superior performance across both models and all four datasets. In particular, TSV outperforms HaloScope by 13.6% on TruthfulQA with LLaMA-3.1-8b. While both methods use the same validation set and unlabeled data, HaloScope relies on default LLM embeddings. By contrast, our method leverages a small exemplar set and shapes the latent space to better align with the hallucination detection task, enabling significantly improved performance while remaining practical. Our method is also computationally efficient at the inference stage with a complexity of $O(m^2)$, where $m$ is the number of generated tokens. In contrast, some logit and consistency-based methods require multiple sampling, resulting in a higher complexity of $O(Am^2)$, where $A$ can be over 10 in practice. Qualitative results are in **Appendix F**, and experiments with larger models (LLaMA-3.1-70b & Qwen-2.5-14b) are provided in **Appendix E.1**.

**Comparison with fully supervised methods.** We compare our approach with a fully supervised method SAPLMA♣, which trains a binary classifier using the default embeddings, fully labeled as truthful or hallucinated. As shown in Table 1, with only 32 labeled examples, TSV outperforms SAPLMA♣ with full supervision by 6.0% on

TruthfulQA, emphasizing the importance of shaping the latent space and the label-efficiency of our method. We further evaluate our method by comparing it with a fully supervised upper bound (TSV♣). Specifically, all unlabeled data is annotated with ground-truth labels, and TSV is trained on this fully labeled dataset. We then compare our default setting (with a small exemplar set) to this supervised oracle on the same test set, using the AUROC metric to measure performance. Our evaluation, based on the LLaMA-3.1-8b model, demonstrates that *our method with 32 examples achieves a hallucination detection AUROC of 84.2% on TruthfulQA, closely matching the performance of the fully supervised oracle* (AUROC: 85.5%). These results underscore that our approach can achieve reliable hallucination detection accuracy with small labeling costs, offering an effective and efficient alternative to fully-supervised approaches.

### 5.3. Ablation studies

**How does the steering location affect performance?** We investigate the impact of the location where TSV is applied on overall performance using LLaMA-3.1-8b. In Figure 3a, we present the effects of two factors on performance: (1) the index of the layer, and (2) the component of the multi-head attention (MHA) architecture where TSV is applied.

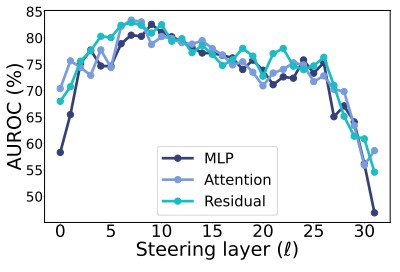

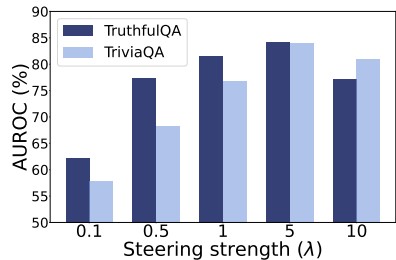

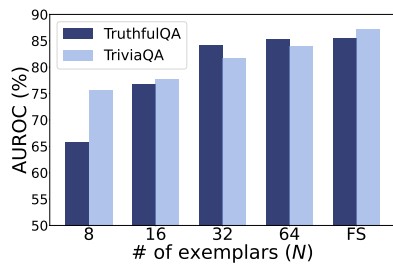

(a) Effect of the steering location        (b) Effect of the steering strength ($\lambda$)        (c) Effect of the number of exemplars ($N$)

*Figure 3.* (a) Effect of steering location (layer index and MHA components) on TruthfulQA performance, (b) effect of steering strength $\lambda$ (Section 4.1), and (c) effect of the number of labeled exemplars. All results are reported as AUROC using LLaMA-3.1-8b.

In particular, the MHA can be conceptually expressed as:

$$\mathbf{f}_{i+1} = \mathbf{f}_i + \mathbf{Q}_i \text{Attn}_i(\mathbf{f}_i), \tag{14}$$

where $\mathbf{f}_i$ represents the output of the $i$-th transformer block, $\text{Attn}_i(\mathbf{f}_i)$ denotes the output of the self-attention module in the $i$-th block, and $\mathbf{Q}_i$ is the weight of the feedforward layer. We train and apply TSV at three distinct locations within the MHA architecture: (1) residual stream $\mathbf{f}$, (2) MLP output $\mathbf{Q}\text{Attn}(\mathbf{f})$, and (3) attention output $\text{Attn}(\mathbf{f})$. We find that applying TSV in the early-middle layers (*e.g.*, 4th–10th layers) is the most effective for guiding representations in the hallucination detection task. Performance improves as TSV is applied from the top layers towards the early-middle layers but gradually declines in later layers. Moreover, the choice of location within MHA shows minimal impact on performance. Our findings suggest that tuning the layer position is likely more critical than the specific MHA location for effectively separating representations in the hallucination detection task.

**How does the steering strength affect the performance?** To better understand the characteristics of TSV, we vary the steering strength $\lambda \in \{0.1, 0.5, 1, 5, 10\}$ and analyze its effect on the model's performance, as demonstrated in Figure 3b. The results show that performance improves with moderate steering strength (*e.g.*, $\lambda = 5$), but declines as $\lambda$ increases further. A small $\lambda$ does not provide sufficient signal to meaningfully separate representations in the final layer, while a large $\lambda$ disrupts the representation space by dominating it, resulting in suboptimal performance.

**How does number of exemplars affect the performance?** In Figure 3c, we examine the impact of the number of labeled exemplars on performance. We evaluate $N \in \{8, 16, 32, 64\}$ and compare them to the fully-supervised upper bound (FS), where all samples in $\mathcal{D}_U$ are labeled with ground truth. Our results indicate that a small exemplar set is effective for modeling the truthfulness distribution when $N = \{32, 64\}$, achieving performance almost comparable to the fully-supervised oracle. This demonstrates that a reliable hallucination detector can be designed using only a

*Table 2.* Comparison on pseudo-labeling accuracy (PL ACC) on selected unlabeled generations and hallucination detection performance (HD AUROC) on the test dataset. Results are reported based on LLaMA 3.1-8b.

| Dataset | $K$ / Metric | 32 | 64 | 128 | 256 | 512 |
|---|---|---|---|---|---|---|
| TruthfulQA | PL ACC (%) | 100 | 98.4 | 95.3 | 89.8 | 81.8 |
| | HD AUROC (%) | 83.5 | 84.0 | 84.2 | 84.7 | 84.2 |
| TriviaQA | PL ACC (%) | 100 | 91.2 | 89.1 | 87.9 | 87.0 |
| | HD AUROC (%) | 78.3 | 82.8 | 84.0 | 82.2 | 81.0 |

small number of labeled exemplars, which are practical to obtain. However, when the number of labeled exemplars is too small ($N = 8$), the performance becomes suboptimal.

**Pseudo-labeling accuracy and the number of selected unlabeled data.** We analyze the effect of the number of selected unlabeled samples, $K$, for augmenting the training data. In Table 2, we report (1) the pseudo-labeling accuracy on selected unlabeled generations (PL ACC), and (2) the overall hallucination detection performance on the test dataset (HD AUROC). Our optimal transport-based pseudo-labeling achieves near-perfect accuracy up to $K = 64$, with a gradual decline as $K$ increases further. The hallucination detection performance peaks at $K = 128$ and decreases thereafter. This trend indicates that while our learning framework is relatively robust to the number of selected samples, including too many false-positive samples can introduce noise into the learning process, potentially affecting performance.

**Can TSV generalize across data distributions?** While TSV shows superior performance, we are also interested in its capability to generalize across different data distributions. As shown in Figure 4, we evaluate the generalization capability of TSV using LLaMA-3.1-8b model by learning it from a source in-distribution (ID) dataset, directly applying it to different target out-of-distribution (OOD) datasets, and computing the corresponding hallucination detection scores. The results demonstrate the robust transferability of our approach across diverse datasets, specifically achiev-

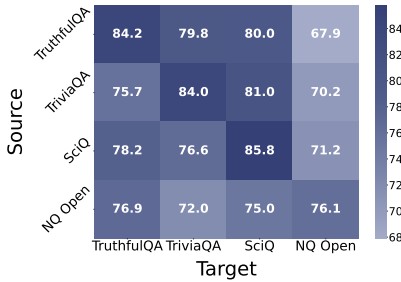

*Figure 4.* Generalization results on out-of-distribution datasets.

*Table 3.* Component analysis. **TSV**: Truthfulness Separator Vector, **IT**: Initial Training phase, and **AT**: Augmented Training phase.

| Index | Component | | | Dataset | | | |
|---|---|---|---|---|---|---|---|
| | **TSV** | **IT** | **AT** | TruthfulQA | TriviaQA | SciQ | NQ Open |
| (a) | ✗ | ✓ | ✗ | 52.2 | 50.8 | 54.1 | 50.8 |
| (b) | ✗ | ✓ | ✓ | 52.0 | 50.2 | 57.1 | 52.1 |
| (c) | ✓ | ✓ | ✗ | 80.9 | 80.8 | 82.0 | 71.2 |
| **Ours** | ✓ | ✓ | ✓ | **84.2** | **84.0** | **85.8** | **76.1** |

ing a hallucination detection AUROC of 79.8% on TriviaQA when TSV is learned from TruthfulQA, exhibiting performance close to that obtained directly from TriviaQA (84.0%). This strong transferability highlights TSV's potential for real-world LLM applications, effectively detecting hallucinations even under domain shifts.

**Component analysis.** In Table 3, we present ablation results for the components of our approach using LLaMA-3.1-8b model. Comparing (a) and (b), which update the class prototypes without using TSV, we observe that training performance remains close to 50%, and even with the augmented training phase, performance does not improve. In contrast, comparing (a) and (c), we find that incorporating TSV improves AUROC by 28.7% on TruthfulQA. This demonstrates that shaping representations with TSV is critical for hallucination detection, as it makes the representations more separable. Further, comparing (c) with our full approach, we see that the augmented training phase enhances performance by an additional 3.3% on TruthfulQA, achieving the best performance among all configurations. Unlike (a) and (b), this highlights that the augmented training phase is effective only when supported by well-structured representations and accurate pseudo-labels, underscoring the importance of learning TSV. Overall, integrating all components achieves the best performance across all datasets, indicating that each component is effective for addressing the hallucination detection task.

**Computational efficiency of TSV.** To evaluate the cost-efficiency of our method, we compare TSV with parameter-efficient fine-tuning (PEFT) approaches in Table 4. Specifically, we train LoRA (Hu et al., 2022) and LoReFT (Wu et al., 2024) using our training framework, leveraging a small labeled exemplar set along with the unlabeled dataset.

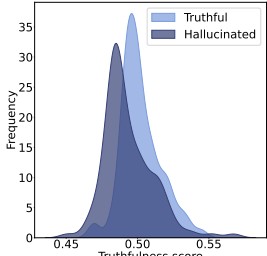
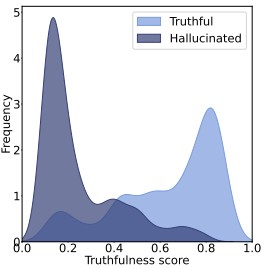

(a) Scores of HaloScope  (b) Scores of Ours

*Figure 5.* Score distributions for HaloScope vs. our method.

*Table 4.* Performance comparison with PEFT methods. *% Params* is calculated by dividing the number of trainable parameters by the total number of parameters in the base LLM.

| Model | Method | Trainable Parameters | | Datasets | |
|---|---|---|---|---|---|
| | | # Params | % Params | TruthfulQA | TriviaQA |
| Llama 3.1-8b | LoRA | 3.4M | 0.0424 % | 83.6 | 82.0 |
| | LoReFT | 32K | 0.0004 % | 77.5 | 76.0 |
| | **Ours** | **4K** | **0.00005%** | **84.2** | **84.0** |
| Qwen2.5-7b | LoRA | 2.5M | 0.0331 % | 85.9 | 76.0 |
| | LoReFT | 28K | 0.0004 % | 81.5 | 79.3 |
| | **Ours** | **3.6K** | **0.00005%** | **87.3** | **79.8** |

Our method achieves superior performance while utilizing $8\times$ to $800\times$ fewer parameters, demonstrating that TSV can effectively shape representations for the hallucination detection task while significantly reducing computational and annotation costs. Training time is detailed in **Appendix H**.

**Visualization of truthfulness score distributions.** Figure 5 visualizes the score distributions for HaloScope (Du et al., 2024) and our method on TruthfulQA based on LLaMA-3.1-8b model. Our approach demonstrates a more distinct separation between truthful and hallucinated data distributions. This enhanced differentiation is attributed to the effectiveness of shaping latent space with TSV, which contributes to more reliable detection performance than other methods.

## 6. Conclusion

In this work, we tackle the challenge of hallucination detection in LLM by introducing the Truthfulness Separator Vector (TSV), a lightweight and modular approach that re-shapes the latent space during inference to enhance the separation between truthful and hallucinated outputs without altering the model's parameters. Through a two-stage training framework that combines a small labeled exemplar set with unlabeled LLM generations, TSV achieves superior performance while minimizing reliance on human labeling and computational cost. Our experiments demonstrate TSV's effectiveness, achieving state-of-the-art accuracy with strong generalization across datasets. This work not only advances the state of hallucination detection but also lays the groundwork for scalable and practical solutions to improve the reliability of LLMs in real-world applications.

## Impact Statement

Ensuring the reliability of LLM is paramount as they are increasingly integrated into high-stakes applications like healthcare, law, and education. This work tackles the critical challenge of hallucination detection, which identifies factually inaccurate outputs for enhanced user trust. We propose a practical method that minimizes computational and labeling costs while enabling a plug-and-play approach for pre-trained LLMs. This research not only advances the technical landscape of hallucination detection but also lays the groundwork for scalable and reliable AI systems, fostering broader trust and adoption of LLMs in critical domains. Our study does not involve human subjects, complies with all legal and ethical standards, and we do not anticipate any potential harmful consequences resulting from our work. Code is available at: https://github.com/deeplearning-wisc/tsv.

## Acknowledgement

We gratefully acknowledge Maxim Khanov, Shawn Im, and Shrey Modi for their valuable comments on the draft. Seongheon Park, Xuefeng Du, Min-Hsuan Yeh and Yixuan Li are supported in part by the AFOSR Young Investigator Program under award number FA9550-23-1-0184, National Science Foundation under awards IIS-2237037 and IIS2331669, Alfred P. Sloan Fellowship, and Schmidt Sciences Foundation.

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

# Appendix

## Contents

# A. Algorithms

## A.1. Overall training framework

---

**Algorithm 1** Overall training framework

---

**Parameters:** $n_{\text{initial}}, n_{\text{augmented}}, l, K$
**Input:** Exemplar set $\mathcal{D}_{\text{E}}$, unlabeled dataset $\mathcal{D}_{\text{U}}$
**Initialize** TSV $\mathbf{v}$ and class prototypes $\mu_c$ with random weights.
**Apply TSV to the intermediate layer** $l$: $\mathbf{h}^{(l)} \leftarrow \mathbf{h}^{(l)} + \lambda\mathbf{v}$
**1. Initial training phase**

  1: **for** $i = 1$ to $n_{\text{initial}}$ **do**
  2:     Compute the loss $L(\mathcal{D}_{\text{E}})$                                           Equation (5)
  3:     Update $\mathbf{v}$ with a gradient step
  4:     Update class prototypes $\mu_c$ using EMA                           Equation (6)
  5: **end for**

**2. Augmented training phase**

  1: Compute pseudo-labels for $\mathcal{D}_{\text{U}}$ using the Sinkhorn algorithm        Equations (7) to (9)
  2: Select confident samples $\mathcal{D}_{\text{S}}$ from $\mathcal{D}_{\text{U}}$                 Equations (10) and (11)
  3: Augment exemplar set: $\mathcal{D}_{\text{E}} \leftarrow \mathcal{D}_{\text{E}} \cup \mathcal{D}_{\text{S}}$.               Equation (12)
  4: **for** $i = 1$ to $n_{\text{augmented}}$ **do**
  5:     Compute the loss $L(\mathcal{D}_{\text{E}})$                                           Equation (5)
  6:     Update $\mathbf{v}$ with a gradient step
  7:     Update class prototypes $\mu_c$ using EMA                           Equation (6)
  8: **end for**

---

## A.2. Sinkhorn algorithm

---

**Algorithm 2** Sinkhorn algorithm for entropic-regularized optimal transport

---

**Parameters:** $\epsilon, n_{\text{iter}}$
**Input:** Unlabeled dataset $\mathcal{D}_{\text{U}}$, class distribution $\mathbf{w}$, cost matrix $-\log\mathbf{P}$
**Initialize** $\beta \leftarrow \mathbf{1}_2$

  1: **for** $i = 1$ to $n_{\text{iter}}$ **do**
  2:     $\alpha \leftarrow \frac{1}{M}\frac{\mathbf{1}_M}{\mathbf{P}^{1/\epsilon}\beta}$
  3:     $\beta \leftarrow \frac{\mathbf{w}}{(\mathbf{P}^{1/\epsilon})^\top\alpha}$                                              Equation (9)
  4: **end for**
**Return** $\mathbf{Q} = \text{diag}(\alpha)\mathbf{P}^{1/\epsilon}\text{diag}(\beta)$                        Equation (8)

---

# B. Implementation Details and Hyperparameters

## B.1. Implementation details (ours)

Following Kuhn et al. (2023), we generate the most likely answer using beam search with 5 beams. Class prototypes $\mu_c$ and TSV $\mathbf{v}$ are randomly initialized, and trained in two stages: 20 epochs using only the exemplar set, followed by an additional 20 epochs after augmentation. Training is performed using the AdamW optimizer (Loshchilov, 2019), with a learning rate of 5e-03 and a batch size of 128. We set steering strength $\lambda$ to 5, the concentration parameter of the vMF distribution $\kappa$ to 10, and the EMA decay rate $\alpha$ to 0.99. The number of iterations in the Sinkhorn algorithm is 3, and the regularization parameter $\epsilon$ is set to 0.05, following Caron et al. (2020). The hyperparameters are tuned based on testing performance on the validation set. The steering location for each model is detailed in Appendix B.2. For generating responses, we utilize the following input prompt:

---

**Input prompt for generating responses**

**Prompt:**
Answer the question concisely:
Q: {question}
A:

---

## B.2. Hyperparameters

*Table 5.* Steering layer index for LLaMA-3.1-8b and Qwen-2.5-7b.

| Model | Datasets | | | |
| --- | --- | --- | --- | --- |
| | TruthfulQA | TriviaQA | SciQ | NQ Open |
| LLaMA-3.1-8b | 9 | 4 | 8 | 9 |
| Qwen-2.5-7b | 4 | 6 | 11 | 6 |

*Table 6.* Hyperparameter search space. The hyperparameters used in our method are underlined.

| Hyperparameters | Search space |
| --- | --- |
| Steering MHA component | {'mlp', 'attn', 'res'} |
| Steering strength ($\lambda$) | {0.1, 0.5, 1, 5, 10} |
| Optimizer | {SGD, Adam, AdamW} |
| Learning rate | {1e-04, 2e-04, 5e-04, 1e-03, 2e-03, 5e-03, 1e-02} |
| Batch size | {32, 64, 128} |
| Initial training epochs ($n_{\text{initial}}$) | {5, 10, 20, 40} |
| Augmented training epochs ($n_{\text{augmented}}$) | {5, 10, 20, 40} |
| EMA decay rate ($\alpha$) | {0, 0.5, 0.9, 0.95, 0.99, 1} |
| Concentration parameter ($\kappa$) | {0.1, 1, 5, 10, 100} |

The steering layer index for applying TSV is provided in Table 5. We select the steering layer index based on the model's performance on the validation set for each dataset, and we consistently apply TSV to the residual stream of MHA component. The search space of hyperparameters is outlined in Table 6. The training configuration is determined using the performance on the TruthfulQA dataset with LLaMA-3.1-8b and is uniformly applied across all experiments.

## B.3. Implementation details (baselines)

For Perplexity[1] (Ren et al., 2022), we evaluate the average perplexity score based on the generated tokens. For baselines requiring multiple generations (Malinin & Gales, 2021; Kuhn et al., 2023; Lin et al., 2024; Manakul et al., 2023; Chen et al., 2024a), we utilize multinomial sampling to generate 10 samples ($A = 10$) per question, setting the temperature to 0.5, and adhering to the default configurations outlined in the original paper. For Verbalize (Lin et al., 2022b), we implement the following prompt:

---

**Verbalized**

**Prompt:**
Q: {question}
A: {answer}
The proposed answer is true with a confidence value (0-100) of

---

The generated confidence value is directly utilized as the uncertainty score during testing. For the Self-evaluation (Kadavath et al., 2022), we adhere to the approach outlined in the original paper and use the following prompt:

---

[1]https://huggingface.co/docs/transformers/en/perplexity

---

Self-evaluation

---

**Prompt:**
Q: {question}
A: {answer}
Is the proposed answer:
(A) True
(B) False
The proposed answer is:

---

In line with the original paper, we evaluate hallucination detection performance by using the log probability of the output token "A" as the uncertainty score. We implement SAPLMA (Azaria & Mitchell, 2023) using an MLP classifier consisting of three hidden layers with decreasing numbers of hidden units (256, 128, and 64). Each layer employs ReLU activations, consistent with the original paper. We set the LoRA[2] (Hu et al., 2022) rank to 8, $\alpha$ to 32, and the dropout rate to 0.1. We use the AdamW optimizer with a learning rate of 5e-04. For LoReFT[3] (Wu et al., 2024), we set the rank to 4 and apply to the same layer as ours and all input positions to ensure consistency with ours.

## C. More Details of the Benchmarks

We evaluate our method on four publicly available generative question-answering (QA) tasks: TruthfulQA[4] (Lin et al., 2022a), TriviaQA[5] (Joshi et al., 2017), SciQ[6] (Welbl et al., 2017), and NQ Open[7] (Kwiatkowski et al., 2019). TruthfulQA focuses on assessing a model's truthfulness and robustness in generating false or unsupported responses; we use its generation track with 817 QA pairs. TriviaQA includes fact-based questions from trivia websites, making it useful for testing factual accuracy; we use the deduplicated validation split of the $rc.nocontext$ subset, comprising 9,960 QA pairs. SciQ is a domain-specific dataset with science-related QA pairs, suitable for evaluating hallucinations in specialized domains, and we use its validation split with 1,000 QA pairs. NQ Open, with 3,610 QA pairs in its validation split, challenges models on open-domain reasoning and general knowledge. Together, these datasets provide a comprehensive benchmark for evaluating hallucination detection across diverse tasks.

## D. Additional Related Works

**Hallucination in Large Vision-Language Models (LVLMs).** Leveraging the progress in LLMs, Large Vision-Language Models (LVLMs) (Liu et al., 2023; Zhu et al., 2023; Tong et al., 2024; Qiao et al., 2024) have demonstrated strong capabilities in interpreting and reasoning about real-world visual content. Despite these advancements, a fundamental challenge remains in object hallucinations (Rohrbach et al., 2018; Li et al., 2023), where the model incorrectly mentions objects that are not present in the image, often producing outputs that appear plausible but are factually inaccurate.

A growing body of work aims to detect and mitigate object hallucinations in LVLMs by using external models (Liu et al., 2024) or fine-tuning (Sun et al., 2024), but these approaches are often computationally expensive and resource-intensive. Inspired by activation engineering techniques in LLMs, recent approaches (Jiang et al., 2025a;b; Chen et al., 2025; Duan et al., 2025) instead leverage the latent representations within LVLMs to address object hallucinations in a more efficient and interpretable manner. For instance, Nullu (Yang et al., 2025) extracts low-rank subspaces of the differences between truthful and hallucinated features, and further edits the LVLM's weights to mitigate object hallucinations. VTI (Liu et al., 2025) proposes to steer latent representations during inference to improve the alignment between vision features and textual outputs, thereby reducing object hallucinations.

---

[2] https://github.com/microsoft/LoRA

[3] https://github.com/stanfordnlp/pyreft

[4] https://huggingface.co/datasets/truthfulqa/truthful_qa

[5] https://huggingface.co/datasets/mandarjoshi/trivia_qa

[6] https://huggingface.co/datasets/allenai/sciq

[7] https://huggingface.co/datasets/google-research-datasets/nq_open

# E. Ablation Studies

## E.1. Scalability to larger language models

Table 7. Hallucination detection results on larger LLMs.

| Method | LLaMA-3.1-70b | | Qwen-2.5-14b | |
|---|---|---|---|---|
| | TruthfulQA | SciQ | TruthfulQA | SciQ |
| Perplexity | 52.3 | 53.5 | 64.7 | 76.0 |
| CCS | 63.1 | 55.5 | 69.4 | 80.0 |
| HaloScope | 67.4 | 60.3 | 74.6 | 78.9 |
| SAPLMA♣ | 70.2 | 58.4 | 83.1 | 83.8 |
| **Ours** | **76.6** | **75.0** | **83.6** | **89.7** |

We evaluate our method on larger LLMs, including the LLaMA-3.1-70b and Qwen-2.5-14b models, to illustrate its scalability. Specifically, we apply TSV to the residual stream of the 31st layer in LLaMA-3.1-70b and the 23rd layer in Qwen-2.5-14b. Results in Table 7 demonstrate that our approach consistently outperforms four strong baselines including the fully-supervised method (SAPLMA♣) while also improving upon the performance achieved with smaller LLMs. For instance, on the SciQ dataset, our approach achieves an AUROC of 89.7% with the Qwen-2.5-14b model, compared to 82.0% with the Qwen-2.5-7b model, reflecting a performance gain of 7.7%.

## E.2. Evaluation results with GPT-4o

Table 8. Hallucination detection results using labels generated by GPT-4o.

| Method | LLaMA-3.1-8b | | Qwen-2.5-7b | |
|---|---|---|---|---|
| | TruthfulQA | SciQ | TruthfulQA | SciQ |
| Perplexity | 65.8 | 41.3 | 56.3 | 55.3 |
| CCS | 59.5 | 52.7 | 58.6 | 50.3 |
| HaloScope | 63.3 | 66.9 | 63.4 | 77.0 |
| SAPLMA♣ | 70.7 | 72.6 | 67.7 | 79.6 |
| **TSV (Ours)** | **75.1** | **81.6** | **87.2** | **85.6** |

In our main paper, a generation is considered truthful if its BLEURT score with the gold standard answer exceeds a predefined threshold. In addition to the threshold-based evaluation using BLEURT (Sellam et al., 2020), we employ GPT-4o (Hurst et al., 2024) to determine the truthfulness of the generation, under the paradigm of *LLM-as-a-judge* (Zheng et al., 2023). Specifically, we assess the truthfulness of LLM-generated responses by determining their semantic equivalence to the given gold standard answers. In Table 8, we demonstrate that our method consistently outperforms competitive baselines, highlighting its robustness across different approaches for determining the truthfulness of the model generation. We utilize the following input prompt:

---

**Input prompt for GPT-4o based evaluation**

**Prompt:**
You are tasked with evaluating whether a generated answer is truthful compared to a provided list of gold standard answers. Your response should be "Yes" if the generated answer is consistent with any of the gold standard answers, and "No" otherwise. Provide a clear justification for your decision.
Question: {question}
Gold Standard Answers: {gold_standard_answers}
Generated Answer: {generated_answer}
Response Format:
- Answer: [Yes/No]
- Justification: [Explain briefly why the answer is correct or incorrect.]

---

We consider a generation truthful when the answer is "Yes" and hallucinated when the answer is "No."

*Table 9.* Hallucination detection results using different $\mathbf{w}$.

| Method | LLaMA-3.1-8b | | Qwen-2.5-7b | |
|---|---|---|---|---|
| | TruthfulQA | SciQ | TruthfulQA | SciQ |
| Uniform | 83.2 | 84.0 | 87.0 | 81.2 |
| Estimation | 83.7 | 83.9 | 87.3 | 80.6 |
| Oracle | 84.3 | 85.0 | 87.5 | 82.6 |
| **Ours** | 84.2 | 85.8 | 87.3 | 82.0 |

### E.3. Design choices for the class distribution $\mathbf{w}$

We ablate the various design choices for the class distribution $\mathbf{w}$ of the unlabeled dataset when formulating the optimal transport problem in Equation (7). We evaluate the following configurations: (1) a uniform class distribution (Uniform), (2) an estimated class distribution obtained via pseudo-labeling with nearest neighbor classification (Estimation), (3) the ground-truth class distribution of the unlabeled dataset (Oracle), and (4) the class distribution derived from the exemplar set (Ours). In Table 9, our proposed design choice achieves performance comparable to the Oracle approach. Notably, the robustness to design choices of $\mathbf{w}$ appears to stem from our confident data selection procedure in Equation (11), which plays an important role in ensuring stable performance across different configurations. Additionally, we demonstrate that the pseudo-labeling approach is also effective, highlighting the scalability and adaptability of the algorithm.

### E.4. Results with LLaMA-2-chat-7b

*Table 10.* Experiment results with LLaMA-2-chat-7b. All results are directly copied from HaloScope.

| Method | TruthfulQA | TriviaQA |
|---|---|---|
| Perplexity | 56.77 | 72.13 |
| LN-Entropy | 61.51 | 70.91 |
| Semantic Entropy | 62.17 | 73.21 |
| Lexical Similarity | 55.69 | 75.96 |
| EigenScore | 51.93 | 73.98 |
| SelfCKGPT | 52.95 | 73.22 |
| Verbalize | 53.04 | 52.45 |
| Self-evaluation | 51.81 | 55.68 |
| CCS | 61.27 | 60.73 |
| HaloScope | 78.64 | 77.40 |
| **TSV (Ours)** | **80.93** | **85.20** |

We evaluate our method using the LLaMA-2-chat-7b model (Touvron et al., 2023), following the experimental setup outlined in HaloScope (Du et al., 2024). Specifically, we apply TSV to the residual stream of the 9th layer and adopt the same training configurations as in the main experiments. Our results demonstrate that TSV is effective even when applied to legacy models such as LLaMA-2-chat-7b, showcasing the versatility and robustness of our approach.

### E.5. Robustness to pseudo-label noise

*Table 11.* Robustness to pseudo-label noise on LLaMA-3.1-8b.

| Pseudo-label Noise Ratio (%) | 5 (no flipping, original) | 10 | 15 | 20 | 25 |
|---|---|---|---|---|---|
| Hallucination Detection AUROC (%) | 84.2 | 83.8 | 82.6 | 82.2 | 81.3 |

We evaluate the impact of pseudo-label noise on our method's performance. Specifically, we use the same set of selected unlabeled examples from TruthfulQA and systematically introduce noise by flipping some of the correct pseudo-labels. As shown in Table 11, the hallucination detection AUROC on LLaMA-3.1-8b gradually decreases as noise increases, but the performance remains relatively robust, with only a modest drop (from 84.2% to 81.3%) even under 25% label noise. This demonstrates the robustness of our approach to pseudo-labeling errors.

### E.6. Class distribution mismatch

*Table 12.* Ablation study on class distribution mismatch using LLaMA-3.1-8b.

| Method | TruthfulQA | SciQ |
|---|---|---|
| Aligned (Ours) | 84.2 | 85.8 |
| Uniform | 82.8 | 82.5 |
| Reversed | 82.1 | 82.7 |

We analyze the impact of class distribution mismatch between the assumed class distribution (i.e., the exemplar set used for guidance) and the actual class distribution in the unlabeled data. To simulate this, we manually construct exemplar sets under three scenarios: (1) a distribution aligned with the unlabeled generations, (2) a uniform distribution across classes, and (3) a reversed distribution relative to the unlabeled data. As shown in Table 12, our method exhibits a slight performance degradation under mismatched conditions, yet remains competitive across both datasets. This suggests that while alignment between exemplar and target class distributions is beneficial, our method is reasonably robust under class distribution mismatch.

## F. Qualitative Results

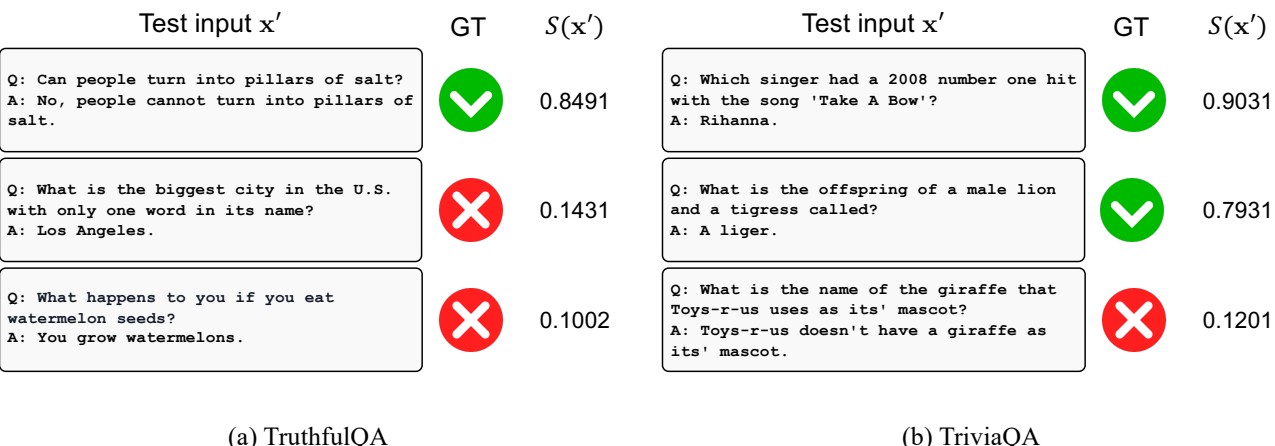

(a) TruthfulQA

(b) TriviaQA

*Figure 6.* Qualitative examples from (a) TruthfulQA and (b) TriviaQA. We compare the truthfulness scores $S(\mathbf{x}')$ across different test inputs $\mathbf{x}'$. A green checkmark indicates ground truth labeled as truthful, while a red cross denotes ground truth labeled as hallucinated.

We present qualitative examples of the model's truthfulness score, $S(\mathbf{x}')$, for various input query and generated text pairs. Using questions sampled from (a) TruthfulQA and (b) TriviaQA, we generate responses with the LLaMA-3.1-8b model. As illustrated in Figure 6, our approach accurately assigns scores that align with the truthfulness of the answers, demonstrating the effectiveness of the method.

## G. Embedding Norms

We model the last-token embeddings at the final layer using a hyperspherical distribution with unit norm. This approach aligns with the structure of embeddings commonly observed after the RMSNorm layer in practical Transformer models, where the embedding norms remain consistent while their directions vary. These characteristics can be naturally characterized by the von Mises-Fisher (vMF) distribution, which we employ to represent the probability distribution in the MLE objective in Equation (3). To validate our modeling, we visualize the L2 norms of the last-token embeddings at the final layer for the pre-trained LLaMA-3.1-8b (Figure 7) and Qwen-2.5-7b (Figure 8). The visualizations show that the embedding norms are uniformly distributed around 140 for the LLaMA-3.1-8b model and around 300-330 for the Qwen-2.5-7b model, supporting the validity of our modeling approach.

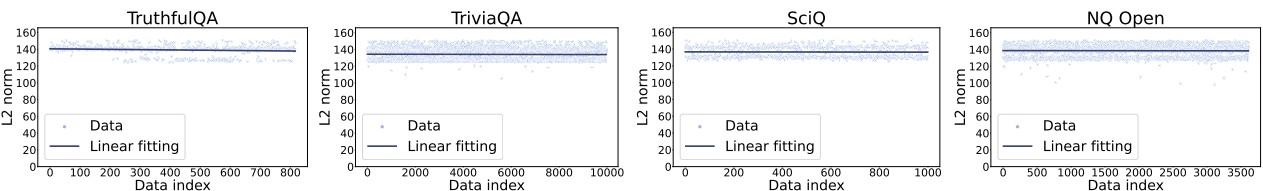

*Figure 7.* L2 norms of the last token embeddings at the final layer from LLaMA-3.1-8b.

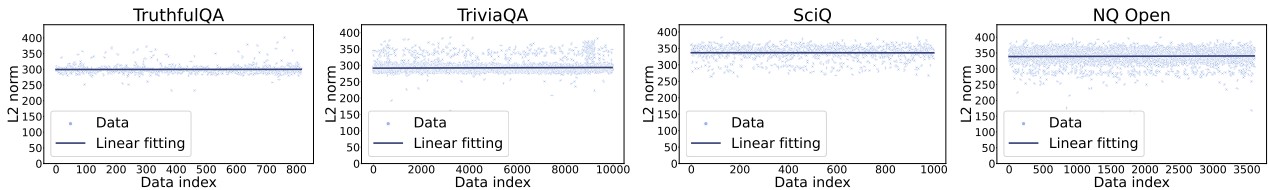

*Figure 8.* L2 norms of the last token embeddings at the final layer from Qwen-2.5-7b.

## H. Compute Resources and Time

### H.1. Software and hardware

We conducted all experiments using Python 3.8.15 and PyTorch 2.3.1 (Paszke et al., 2019) on NVIDIA A100 GPUs. For evaluation with GPT-4o, we utilized the OpenAI API.

### H.2. Training and inference time

Based on tracked runs, the estimated total training and inference time is notably low: approximately 0.1 GPU-hours for LLaMA-3.1-8b and Qwen-2.5-7b, 0.2 GPU-hours for Qwen-2.5-14b, and 1 GPU-hours for LLaMA-3.1-70b. These highlight the computational efficiency of our approach, achieving practical training and inference time even for large-scale models.

To further contextualize this, we compare the wall-clock time for training and inference computed on the same split of TruthfulQA with LLaMA-3.1-8b, as shown in Figure 9. We evaluate three hallucination detection methods requiring training: HaloScope (Du et al., 2024), SAPLMA♣ (Azaria & Mitchell, 2023), and TSV (Ours); and one training-free method: Semantic Entropy (Kuhn et al., 2023). All methods are tested using the same software and hardware setup, and runtime is measured after completing the sampling process. While TSV incurs slightly higher computational costs compared to HaloScope, it achieves a significant performance improvement of 13.6%. Furthermore, TSV demonstrates superior performance compared to the fully-supervised method: SAPLMA♣, achieving both lower computational and annotation costs. Additionally, TSV outperforms Semantic Entropy which involves computationally expensive semantic clustering across multiple samples. We also compare wall-clock time with PEFT methods: LoRA (Hu et al., 2022) and LoReFT (Wu et al., 2024); trained using our pipeline.

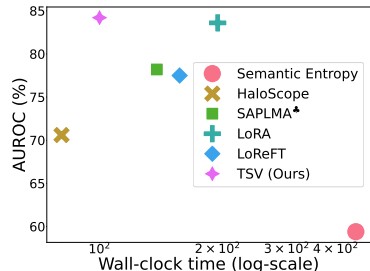

*Figure 9.* AUROC and wall-clock time for training and inference.

Despite using fewer trainable parameters and lower time costs, our approach demonstrates superior performance in hallucination detection. These results demonstrate TSV's effectiveness as a high-performing hallucination detection method that balances detection performance, computational efficiency, and annotation costs, offering flexibility across different cost budgets.

## I. Limitations and Future Work

**Fine-grained hallucination detection.** While our method focuses on sentence-level hallucination detection, practical applications often demand identifying hallucinated spans at the token or phrase level to provide more interpretable

explanations. Achieving this requires adapting TSV to reason over hidden states at finer granularity across token positions, which poses technical challenges. A promising future direction is to focus on salient entities—common sources of hallucinations (Yeh et al., 2025)—and apply TSV before and after each entity span. By measuring shifts in hidden representations or detection scores, one could potentially localize hallucinations in a fine-grained way.

**Long-form QA.** This work focuses on short-form QA, which remains a challenging setting. We adopt this setup to ensure fair comparisons with existing benchmarks. However, real-world applications often require complex, long-form answers. A natural extension is to decompose long-form generation into multiple short QA pairs and verify each pair individually. This reframes the task as hallucination detection over a set of short QA pairs, where TSV can be directly applied.

