# OpenReview forum: "Steer LLM Latents for Hallucination Detection"
_ICML.cc/2025/Conference — ICML 2025 poster_

### Official Review · Reviewer_EMXm · 2025-03-06

**Overall Recommendation:** 4

**Summary:**

The paper proposes a method named Truthfulness Separator Vector (TSV) to detect the hallucinations in LLMs. The TSV is a lightweight vector that reshapes the LLM's latent space during inference without altering model parameters. A two-stage training framework is used to explore TSV in a similar manner to a semi-supervised manner, which makes sense to me. Extensive experiments demonstrate that TSV performs well with minimal labeled data, exhibiting strong generalization across datasets.

------------------------After rebuttal--------------

Most of my concerns are well addressed, and therefore, I change my score to 4.
However, I still think the steering method mentioned in A5 is worth trying, as it would further strengthen this paper's contributions to the community.

**Claims And Evidence:**

Supporting by pervious study in Haloscope [NeurIPS'24], I think the two-stage method to explore the truthfulness separator vector can work in hallucination detection tasks.

**Essential References Not Discussed:**

I think most of the listed related works are quite essential for understanding the contributions of this paper.

While there are some other works in Large vision language models (LVLMs) that seem to use a similar approach of studying latent features in LVLM between positive and negative samples to adjust the model's behavior:
e.g.

[1] Reducing hallucinations in vision-language models via latent space steering, 2024.

[2] Nullu: Mitigating Object Hallucinations in Large Vision-Language Models via HalluSpace Projection, 2024.

These papers are only on arXiv, from what it seems, but I still suggest searching for related papers using feature steering in hallucination-related issues in LVLM fields.

**Experimental Designs Or Analyses:**

I think most of the experimental designs are sound and valid. While I think it will be better to discuss and briefly evaluate different methods for pseudo-label assignments. The current experiments seem to lack a proper baseline model.

**Methods And Evaluation Criteria:**

Overall, the paper is well-written and easy to follow. The proposed hallucination detection pipeline makes sense. In all the evaluated datasets, the truthful and untruthful latent representations can all be separated well. Are there failure cases when the latent representations overlap? In other words, does the separation assumption for truthful and untruthful features always hold?

**Other Comments Or Suggestions:**

See previous parts.

**Other Strengths And Weaknesses:**

I suggest that the authors further strengthen the argument that using label assignment via optimal transport is necessary for the proposed method, which is a key contribution of this paper. Generally, there can be a series of techniques that can achieve the label assignment for TSV. Why can using other methods hinder the overall detection performance?

Providing a brief discussion will make it easier for readers to make the main contribution.

**Questions For Authors:**

Does the learned TSV be used to steer activation for hallucination mitigation, as in [1]?

[1] Inference-time intervention: Eliciting truthful answers from a language model.

**Relation To Broader Scientific Literature:**

This research intersects with the broader scientific literature on LLM reliability. It builds on existing hallucination detection methods by addressing the limitation of default embeddings. Demonstrating TSV's effectiveness across datasets validates the approach of steering latent spaces, providing a new direction for improving LLM trustworthiness in real-world applications.

**Theoretical Claims:**

The $\epsilon$ in (7) seems not be defined.

---

> ### Author Rebuttal · Authors · 2025-03-27
>
> We greatly appreciate your thoughtful and valuable comments. Below, we provide detailed responses to each of your questions and comments.
>
> > A1. Optimal separability of the representation
>
> Thank you for your insightful question! We acknowledge that there can indeed be failure cases where the representations of truthful and hallucinated contents overlap. For instance, in Table 1, the performance on the NQ Open dataset (76.1%) is relatively lower compared to other datasets, suggesting weaker separability. This is likely due to the inherently challenging QA task, which can make perfect separation harder.
>
> **We'd like to emphasize that our goal is not to guarantee perfect separability, but rather to improve the separability relative to the pre-trained model.** _This improvement holds consistently across all datasets_, even if the final representation space is not perfectly disentangled. As a result, our method enables more reliable deployment compared to prior approaches that rely on fixed pre-trained embeddings.
>
> > A2. $\epsilon$ in Eqauation (7)
>
> The value of $\epsilon$ is defined in **Appendix  L602** where it is set to 0.05, following [1] that demonstrated its effectiveness. For improved clarity, we will move this detail to the main paper. Thank you for pointing this out!
>
>  [1] Caron et al., "Unsupervised Learning of Visual Features by Contrasting Cluster Assignments," NeurIPS 2020
>
> > A3. Different pseudo-label assignment strategies
>
> We appreciate the reviewer for highlighting this important point.
>
> Our motivation for introducing the optimal transport (OT) is that we aim to generate pseudo-labels that align with the class distribution of unlabeled LLM generations, which is inherently **imbalanced**. Traditional pseudo-labeling methods [2,3] can suffer from class imbalance, leading to biased predictions—either toward majority classes or, in some cases, unstable predictions favoring minority classes. This leads to a **mismatch between the class distribution of pseudo-labels and the true underlying distribution**, which can degrade both pseudo-label quality and overall detection performance.
>
> In contrast, our OT-based clustering approach introduces cluster-wise regularization through a constrained optimization framework, **ensuring that cluster sizes align with the underlying or estimated class distribution**.
>
> In comparison, we evaluate two pseudo-labeling baselines—nearest centroid [2] and confidence thresholding [3] on LLaMA-3.1-8b using AUROC. We will include these comparisons and discussions in the main paper.
>
> | Method  | TruthfulQA | SciQ |
> |----|------------|------------|
> |Nearest centroid [2]|    82.2   | 81.8 |
> | Confidence thresholding [3]|    83.1   | 81.6
> | **Ours** |  **84.2**       |    **85.8**  |
>
>
>  [2] Rebuffi et al., "iCaRL: Incremental Classifier and Representation Learning," CVPR 2017
>
>  [3] Sohn et al., "FixMatch: Simplifying Semi-Supervised Learning with Consistency and Confidence," NeurIPS 2020
>
>
> > A4. References on LVLM latent space steering
>
> Thank you for highlighting these interesting works in the LVLM field! We agree that comprehensively studying latent feature steering in LVLMs is also crucial, offering valuable insights that can inform both communities and help shape future directions. We are happy to include the following relevant works from the LVLM literature. We will also ensure a more thorough literature review to identify any additional related studies.
>
> [4] Liu et al., "Reducing Hallucinations in Vision-Language Models via Latent Space Steering," ICLR 2025
>
> [5] Yang et al., "Nullu: Mitigating Object Hallucinations in Large Vision-Language Models via HalluSpace Projection," CVPR 2025
>
> [6] Chen et al., "ICT: Image-Object Cross-Level Trusted Intervention for Mitigating Object
> Hallucination in Large Vision-Language Models," CVPR 2025
>
> [7] Duan et al., "TruthPrInt: Mitigating LVLM Object Hallucination Via Latent Truthful-Guided Pre-Intervention," ArXiv 2025
>
>
> > A5. Can TSV mitigate hallucination?
>
> Thank you for the interesting question. TSV is designed to learn a representation space specifically for detection (i.e., classification), which is fundamentally different from mitigation (i.e., generation). As a result, it is not straightforward to directly apply a detection-trained TSV to a mitigation task.
>
> However, TSV offers a plug-and-play mechanism for pre-trained LLMs. In practice, one can first apply TSV to classify truthful vs. hallucinated outputs from unlabeled generations in the wild, and then leverage these predictions to learn a steering vector for the mitigation task [8]. **This perspective opens up a promising direction for hallucination mitigation under unsupervised/limited-label settings, where extensive supervision is often impractical**.
>
> [8] Li et al., "Inference-Time Intervention: Eliciting Truthful Answers from a Language Model," NeurIPS 2023

---

### Official Review · Reviewer_tv5M · 2025-03-13

**Overall Recommendation:** 3

**Summary:**

This paper introduces the Truthfulness Separator Vector (TSV), a lightweight approach for hallucination detection in LLMs that reshapes the model's latent space during inference without modifying its parameters. The method employs a single trainable vector added to an intermediate layer, trained through a two-stage framework that first uses a small labeled exemplar set (as few as 32 examples) and then incorporates pseudo-labeled unlabeled data using an optimal transport algorithm with confidence-based filtering. Experiments across multiple datasets (TruthfulQA, TriviaQA, SciQ, NQ Open) demonstrate state-of-the-art performance, achieving +12.8% AUROC improvement on TruthfulQA while requiring only 0.00005% of model parameters (4K for LLaMA-3.1-8B). The approach shows strong generalization across datasets and model families (LLaMA and Qwen) at various scales (7B-70B parameters), performing comparably to fully supervised methods with significantly less labeled data and 8-800× fewer parameters than parameter-efficient fine-tuning alternatives.

## update after rebuttal
Most of my concerns have been addressed. I will remain my positive score.

**Claims And Evidence:**

The primary claims are well-supported by empirical evidence:
- Superior performance claim: The authors demonstrate substantial improvements over state-of-the-art methods across multiple datasets. - The reported AUROC improvements (e.g., +12.8% on TruthfulQA compared to previous methods) are significant and convincing.
- Minimal labeling requirement claim: The paper shows that their approach performs nearly as well with just 32 labeled examples as with full supervision. The ablation studies in Figure 3c provide sufficient evidence for this claim.
- Lightweight and flexible design claim: The comparison with PEFT methods in Table 4 clearly demonstrates TSV's parameter efficiency, using 8-800× fewer parameters than alternatives while achieving superior performance.

**Essential References Not Discussed:**

N/A

**Experimental Designs Or Analyses:**

The experimental designs are generally sound.

**Methods And Evaluation Criteria:**

The proposed methods are appropriate for the hallucination detection problem.

**Other Comments Or Suggestions:**

Consider explaining in more detail how the method might be deployed in practice alongside a production LLM system.

**Other Strengths And Weaknesses:**

Strengths:
- The paper addresses a critical problem in LLM deployment with a pragmatic approach that doesn't require model fine-tuning.
- The component analysis provides valuable insights into the contribution of each part of the system.

Weakness:
- The paper doesn't fully explore potential limitations of the approach, such as its behavior with longer-form content beyond QA pairs.
- While the method shows good generalization across datasets, there's limited discussion of how it might behave across different model architectures or scales beyond the tested models.

**Questions For Authors:**

1. How might the TSV approach scale to much longer generations beyond QA pairs? Many real-world hallucinations occur in longer contexts where there might be a mix of truthful and hallucinated content within a single generation.
2. Have you explored whether the TSV learned from one model architecture (e.g., LLaMA) transfers to a different architecture (e.g., Qwen)? This would strengthen the claim about generalization capabilities.
3. The optimal transport algorithm assumes a class distribution based on the exemplar set. How sensitive is the method to mismatches between the assumed and actual class distributions in real-world data?
4. Could the TSV approach be extended to finer-grained hallucination detection (e.g., identifying specific hallucinated spans within otherwise truthful text)? This would significantly increase its practical utility.
5. The results in Table 4 show TSV outperforming LoRA despite using far fewer parameters. Is this advantage primarily due to the low-data setting, or is there something inherently more effective about the TSV approach for hallucination detection? Would LoRA perform better if updates were restricted to specific components (e.g., only the o_proj weight in the 4th layer) rather than applied broadly across the model?
6. The ablation studies examine different locations for applying TSV, but have you explored applying TSV to multiple locations simultaneously? For example, can the vector be added to multiple positions within the MHA architecture or to multiple layers? This might further enhance the separation capability.
7. How is performance affected when using quantized models (e.g., 4-bit or 8-bit quantization) compared to fp16 models?

**Relation To Broader Scientific Literature:**

N/A

**Theoretical Claims:**

N/A

---

> ### Author Rebuttal · Authors · 2025-03-30
>
> We sincerely appreciate your valuable feedback and insightful questions. Below, we provide detailed responses to each of your points. All the added experiments are performed with LLaMA-3.1-8b.
>
> >A1. Longer generations
>
> Great point. In this work, we focus on short-form QA (phrase and sentence-level) which remains challenging, as evidenced by suboptimal performance reported in prior literature. We adopt this setting to ensure fair comparison with existing benchmark studies. That said, we fully agree that extending to long-form generation is a critical direction for real-world applications. We view our current work as a necessary step toward this goal.
>
> One natural solution is to decompose the long-form generation into multiple QA pairs and verify each pair individually. This reframes the problem as hallucination detection over a set of QA pairs, where TSV can be applied. We believe this warrants a deeper investigation on its own, and we appreciate you highlighting it!
>
>
> > A2. Deployment in practice
>
> TSV is a lightweight, plug-and-play hallucination detection module that can be integrated with any production LLMs. We first perform a standard forward pass using the original model parameters (without TSV) to generate a response. Then, during a second pass, we apply TSV at an intermediate layer and compute a truthfulness score using the steered representation. It can be flexibly toggled on/off, preserving the original model behavior when hallucination detection is not required. We will clarify these deployment steps more explicitly in the revision.
>
> > A3. Class distribution mismatch
>
> Thank you for bringing up this point! We manually construct exemplar sets under three different scenarios: (1) distribution aligned with the unlabeled generations, (2) uniform distribution, and (3) distribution reversed from the unlabeled generations. We observe a slight performance decline in scenarios with distribution mismatches, but the overall performance remains competitive.
>
> |Setting|TruthfulQA|SciQ|
> |-|-|-|
> |Aligned (Ours)|84.2|85.8|
> |Uniform|82.8|82.5|
> |Reversed|82.1|82.7|
>
> > A4. Extension to the span-level hallucination detection
>
> Great question! We agree this is an exciting direction, which requires localizing exactly which tokens or phrases are hallucinated. This is particularly challenging, as it requires adapting TSV to separate hidden states at different token positions.
>
> One promising direction is to identify salient entities—often the primary source of hallucinations—and apply TSV before and after each entity span, then measure the resulting shift in hidden states or detection scores to infer potential hallucinations at the corresponding span. These challenges require deeper investigation. But again, we agree this could significantly increase practical utility and will consider it in future work.
>
>
> > A5. Ablation on LoRA
>
>
> TSV is more effective for hallucination detection as it directly shapes the representation space while being more parameter-efficient. Additionally, we applied selective LoRA updates to specific model components within the same layer as TSV, and found these targeted adjustments still underperformed TSV, as shown below.
>
> |Method|TruthfulQA|SciQ|
> |-|-|-|
> |LoRA (q_proj) |72.0|72.2|
> |LoRA (k_proj) |71.6|74.1|
> |LoRA (v_proj) |76.7|76.8|
> |LoRA (o_proj) |75.2|72.4|
> |**Ours**|**84.2**|**85.8**|
>
> > A6. Applying TSV to multiple locations
>
> We appreciate your insights. We applied TSV to multiple layers or MHA components of LLaMA-3.1-8b on TruthfulQA and observed slight performance gains. However, a single-layer TSV already performs strongly, suggesting it is sufficient for effective hallucination detection and highlights the efficiency of our approach.
>
> |Layer|AUROC|MHA Components|AUROC|
> |-|-|-|-|
> |7, 13|84.2|Res, MLP |82.1|
> |8, 14|84.9|Res, Attn |83.3|
> |9, 12|84.7|MLP, Attn |85.2|
> |Ours (9th layer)|84.2|Ours (Residual)|84.2|
>
> > A7. Performance on the quantized model
>
> Thank you for the great question. We experiment with an 8-bit quantized LLaMA-3.1-8b model$^1$ and observe a slight drop in performance. However, the overall results remain strong, indicating that our method is robust to quantization.
>
> |Model|TruthfulQA|SciQ|
> |-|-|-|
> |8-bit|84.1|84.9|
> |16-bit|84.2|85.8|
>
> $^1$ https://huggingface.co/docs/transformers/main/en/quantization/bitsandbytes
>
>
> > A8. Transferability for different models
>
> Thank you for the suggestion! While we demonstrate that TSV performs well across various models and scales, we believe full transferability is unlikely. This is because differences in architecture, training framework, and pretraining data across models (e.g., LLaMA vs. Qwen) lead to **fundamentally different representation spaces**, making direct application of TSV challenging. However, a promising future direction could be to train an adapter that aligns the representations between models of different architectures or scales.

---

### Official Review · Reviewer_iT9L · 2025-03-14

**Overall Recommendation:** 3

**Summary:**

The paper introduces the Truthfulness Separator Vector (TSV), a lightweight and flexible steering vector designed to reshape the latent space of Large Language Models (LLMs) during inference to enhance the separation between truthful and hallucinated outputs without modifying model parameters. TSV is trained using a two-stage framework that first learns from a small set of labeled exemplars and then augments this set with pseudo-labeled LLM generations using an optimal transport-based algorithm.

**Claims And Evidence:**

N/A

**Essential References Not Discussed:**

N/A

**Experimental Designs Or Analyses:**

See above.

**Methods And Evaluation Criteria:**

Strength:

1. The paper presents a novel and lightweight steering vector method that enhances hallucination detection in LLMs without modifying model parameters.

2. Unlike prior work that relies on costly fine-tuning or extensive labeled datasets, TSV reshapes the latent space dynamically during inference while maintaining model flexibility.

3. The paper provides rigorous experimental validation, demonstrating state-of-the-art hallucination detection on multiple benchmark datasets.

Weakness:

1. A trainable steering vector is not a novel concept for LLMs. The main idea of TSV is to model the binary classification problem of hallucination detection using the von Mises-Fisher distribution and then treat the mean direction of each cluster as a trainable steering vector. While the overall performance is promising, this approach essentially combines existing methods in a new problem setting rather than introducing a fundamentally new technique.

2. The paper primarily uses AUROC as the evaluation metric for hallucination detection. Could you also provide precision (both positive and negative classes), recall, and F1 scores at either the generation level (non-hallucination vs. hallucination) or the token level?

3. TSV heavily relies on the quality of pseudo labels. It would be helpful to include a failure case analysis and assess its robustness to label noise. Additionally, how is the quality of the exemplar set evaluated, and how does the content of exemplars impact the final results?

4. Although TSV avoids fine-tuning, it still requires hyperparameter tuning, which to some extent reduces its practicality in plug-and-play scenarios across diverse models.

**Other Comments Or Suggestions:**

## update after rebuttal

Most of my concerns have been addressed. I would like to keep the current positive score for this work.

**Other Strengths And Weaknesses:**

N/A

**Questions For Authors:**

N/A

**Relation To Broader Scientific Literature:**

N/A

**Theoretical Claims:**

N/A

---

> ### Author Rebuttal · Authors · 2025-03-27
>
> > A1. Steering vector for LLMs
>
> Thank you for your insights! While it is true that the steering vectors have been explored in LLMs, our key contribution lies in how we specifically design them for hallucination detection, as recognized by you and Reviewers iT9L and tv5M. Sections 4.2 and 4.3 illustrate the novel algorithm that leverages both a few labeled and unlabeled LLM generations with an optimal transport-based pseudo-labeling framework for training. To our knowledge, this is the **first work** to formulate hallucination detection in this way.
>
> Moreover, we emphasize that **our contribution lies not only in the specific method but also in the perspective shift we offer to the hallucination detection community**, which has largely relied on fixed pre-trained embeddings—despite their limitations. Our work challenges this status quo by demonstrating that actively steering the latent space—even with a lightweight vector—can substantially improve detection performance. We believe this perspective shift is **fundamentally significant, as it opens up a new design space** and encourages the community to look beyond pre-trained embeddings toward representation shaping as a core principle.
>
> > A2. Evaluation with other metrics
>
> Thank you for the suggestion! In addition to AUROC, we report AUPRC, F1 score, precision, and recall at the generation level, consistent with prior works. We include results on TruthfulQA with LLaMA-3.1-8b, where we found that **our method maintains a consistent advantage across all metrics**.
>
> | Method  | AUROC | AUPRC|  F1 | Precision (Pos) | Precision (Neg) | Recall |
> |-|--|-|-|-|-|-|
> | HaloScope  | 70.2       | 56.2 |64.0  | 55.6| 78.2| 75.3|
> | SAPLMA |  78.2       | 63.3  |64.7       |  57.7| 85.8| 73.4|
> | **Ours** |  **84.2**       | **76.2**  | **70.9**       | **64.5**|**88.5**| **78.9**
>
> > A3-1. Robustness to pseudo-label noise
>
> Thank you for bringing this up! As requested, we include the robustness analysis of pseudo-label noise for the selected unlabeled data ($K=128$). Specifically, with the same selected dataset on TruthfulQA, we gradually add more noise to their pseudo labels by manually flipping some of the correct labels. The resulting relationship between the pseudo-label noise ratio and the hallucination detection AUROC on LLaMA-3.1-8b is shown as follows, where our method remains relatively robust under increasing noise conditions.
>
> | Pseudo-label Noise Ratio (%)  |5 (no flipping, original)  |10 |15 | 20 | 25|
> |-|-|-|-|-|-|
> | Hallucination Detection AUROC (%)| 84.2 |83.8| 82.6 | 82.2 | 81.3|
>
> > A3-2. Quality of exemplar set, and how does the content of exemplars impact the results?
>
> Another insightful point! To understand the quality, we randomly sample the exemplar set using different seeds and annotate them w.r.t. factuality. Manual inspection confirms that the exemplars are high-quality and cover diverse content. The TruthfulQA dataset contains 817 questions across 38 categories with an imbalanced distribution, making it a meaningful testbed for examining how content affects performance. To further verify, we construct exemplar sets on LLaMA-3.1-8b using three strategies, with imbalance ratio $\gamma$ defined as the ratio of the most to least frequent category counts.
>
> | Setting  | AUROC | AUPRC|  F1 |
> |-|-|-|-|
> | Uniform ($\gamma=1$) |     84.9    | 77.0 |  71.0    |
> | Random (Ours, $\gamma=5$) |   84.2  | 76.2 | 70.9 |
> | Biased ($\gamma=10$)|   83.7    | 75.0 |  69.3    |
>
>  **Even in cases where the selected exemplars are biased toward a particular content, the test performance remains consistent.** This reinforces the robustness of our approach.
>
> We emphasize that our approach achieves strong performance with a very small labeled exemplar set (e.g., 32), **which makes manual curation and quality control highly feasible in practice**. Since the cost of labeling such a small set is minimal, our work offers a reliable and efficient solution for hallucination detection in real-world applications.
>
> >A4. Hyperparameters
>
> We emphasize that TSV requires minimal hyperparameter tuning, making it practical and easy to deploy. Key hyperparameters (e.g., steering strength) are **lightweight to tune** and we provide default settings (see Table 6 in Appendix) that **generalize well across all models and datasets**.
>
> In fact, all experiments in our paper use hyperparameters selected using the validation set of TruthfulQA with LLaMA-3.1-8b, and these settings are **applied uniformly without re-tuning to all experiments**, while still achieving state-of-the-art performance.
>
> Moreover, our ablation in Figure 3b demonstrates that TSV’s performance is **not overly sensitive** to the choice of hyperparameters. The robustness of TSV is further supported by our generalization results in Figure 4, which show consistent performance across diverse data distributions, even with fixed hyperparameters. This significantly reduces the burden of hyperparameter tuning in real-world use.

---

### Decision · Program_Chairs · 2025-05-01

**Decision:**

Accept (poster)

**Comment:**

This paper presents a lightweight approach for LLM hallucination detection which reshapes the model's latent space during inference without modifying its parameters. Reviewers acknowledged the contribution and strong performance of the proposed method, while initially raised concerns regarding its difference to prior work and missing important experiments.

After the rebuttal, the authors addressed most of the concerns, and all reviewers agreed to accept this paper. AC read all the reviews, author rebuttals, and the paper, and believes this is a strong paper and recommends for acceptance.